# Sleep Stage Transitions and Sleep-Dependent Memory Consolidation in Children with Narcolepsy–Cataplexy

**DOI:** 10.3390/children10101702

**Published:** 2023-10-18

**Authors:** Katia Gagnon, Amandine E. Rey, Anne Guignard-Perret, Aurore Guyon, Eve Reynaud, Vania Herbillon, Jean-Marc Lina, Julie Carrier, Patricia Franco, Stéphanie Mazza

**Affiliations:** 1Université Claude Bernard Lyon 1, CNRS, INSERM, Centre de Recherche en Neurosciences de Lyon CRNL U1028 UMR5292, FORGETTING, F-69500 Bron, France; katia.gagnon@umontreal.ca (K.G.); amandine.rey@univ-lyon1.fr (A.E.R.); eve.reynaud@univ-lyon1.fr (E.R.); 2National Reference Center for Narcolepsy in the Service of Epilepsy, Sleep and Neuropediatric Functional Explorations of the Woman Mother Child Hospital of Bron, 59, bd Pinel, F-69677 Bron, France; anne.guignard-perret@chu-lyon.fr (A.G.-P.); aurore.guyon@chu-lyon.fr (A.G.); vania.herbillon@chu-lyon.fr (V.H.); patricia.franco@univ-lyon1.fr (P.F.); 3Université Claude Bernard Lyon 1, CNRS, INSERM, Centre de Recherche en Neurosciences de Lyon CRNL U1028 UMR5292, WAKING, F-69500 Bron, France; 4Université Claude Bernard Lyon 1, CNRS, INSERM, Centre de Recherche en Neurosciences de Lyon CRNL U1028 UMR5292, EDUWELL, F-69500 Bron, France; 5Department of Electrical Engineering, École de Technologie Supérieure, Montréal, QC H3C 1K3, Canada; jean-marc.lina@etsmtl.ca; 6Department of Psychology, Université de Montréal, Montréal, QC H3C 3J7, Canada; julie.carrier.1@umontreal.ca

**Keywords:** sleep, children, narcolepsy, sleep transition, memory consolidation, pediatric

## Abstract

Electroencephalographic sleep stage transitions and altered first REM sleep period transitions have been identified as biomarkers of type 1 narcolepsy in adults, but not in children. Studies on memory complaints in narcolepsy have not yet investigated sleep-dependent memory consolidation. We aimed to explore stage transitions; more specifically altered REM sleep transition and its relationship with sleep-dependent memory consolidation in children with narcolepsy. Twenty-one children with narcolepsy–cataplexy and twenty-three healthy control children completed overnight polysomnography and sleep-dependent memory consolidation tests. Overnight transition rates (number of transitions per hour), global relative transition frequencies (number of transitions between a stage and all other stages/total number of transitions × 100), overnight transitions to REM sleep (transition from a given stage to REM/total REM transitions × 100), and altered first REM sleep period transitions (transitions from wake or N1 to the first REM period) were computed. Narcoleptic children had a significantly higher overnight transition rate with a higher global relative transition frequencies to wake. A lower sleep-dependent memory consolidation score found in children with narcolepsy was associated with a higher overnight transition frequency. As observed in narcoleptic adults, 90.48% of narcoleptic children exhibited an altered first REM sleep transition. As in adults, the altered sleep stage transition is also present in children with narcolepsy–cataplexy, and a higher transition rate could have an impact on sleep-dependent memory consolidation. These potential biomarkers could help diagnose type 1 narcolepsy in children more quickly; however, further studies with larger cohorts, including of those with type 2 narcolepsy and hypersomnia, are needed.

## 1. Introduction

Narcolepsy is a lifelong neurological sleep disorder characterized by the instability of sleep–wake regulation and excessive daytime sleepiness, cataplexy, hypnagogic hallucination, sleep paralysis, and nocturnal sleep disruption [1,2]. Narcolepsy is a rare condition whose symptoms start before 18 years of age. The prevalence is estimated to be between 20 and 50 per 100,000; however, the condition is often underdiagnosed, and delays of 5–10 years are common before a firm diagnosis is conducted [3,4,5,6]. There are two types of narcolepsies: narcolepsy type 1 (NT1, narcolepsy with cataplexy and hypocretin deficiency) and narcolepsy type 2 (NT2, formerly narcolepsy without cataplexy). NT1 is caused by a loss of hypocretin-1 neurons located in the dorso-lateral hypothalamus [7], probably via an autoimmune mechanism [8]. Indeed, the human leukocyte antigen (HLA) DQB1*0602 genotype is closely associated with NT1 [9]. Parents of children with narcolepsy frequently report poor school performance, depressed feelings, and impaired family and social life [10,11,12]. A survey carried out in 2020 on patients with narcolepsy highlighted the significant impact of this condition on their daily lives due to, for instance, more academic problems and complaints of attentional deficits [13].

The gold standard for the diagnosis of NT1 is the cerebrospinal fluid hypocretin-1 level [14,15]. However, the lumbar puncture needed to assess hypocretin-1 level is an invasive procedure with a certain degree of risk for children. A mean sleep latency of ≤8 min and the presence of REM periods occurring within 15 min after the sleep onset (SOREMPs) at least twice when falling asleep at night and during multiple sleep latencies tests (MSLT) are currently used as a proxy for the diagnosis of narcolepsy. However, these electroencephalographic (EEG) measures did not show the expected sensitivity and specificity [16,17]. Sorensen and colleagues addressed sleep stage transitions in adults with narcolepsy and disclosed that nocturnal sleep instability (i.e., fluctuations between sleep stages and wakefulness) was associated with biochemical and clinical evidence of hypocretin deficiency and cataplexy [18]. Similar results were observed in orexin knockout mice that have less distinct and more labile states of sleep and wakefulness [19]. Pizza et al. [20] showed that sleep dynamics and more specifically stage transitions reliably identified hypocretin-deficient NT1 in contrast to other central nervous system disorders associated with hypersomnolence. However, in healthy subjects, REM sleep is often preceded by a brief transition into stage 2 of NREM (N2), a direct transition from stage 1 of NREM (N1) or wake (W) to SOREMP is more specific to NT1 and is associated with overall higher narcolepsy severity [21,22,23]. Liu et al. [24] suggested that the altered transition from W or N1 to REM sleep during the first period of REM sleep (FREMP) could be a biomarker and a stable trait associated with the human leukocyte antigen typing (HLA-DQB1*0602) among adults. These findings among adults should be tested in children to provide new diagnostic tools for NT1 and also to understand how sleep instability could impact sleep-dependent memory consolidation, which is rarely explored in children with narcolepsy.

While their intellectual efficiency meets and occasionally exceeds the standard [25], children with narcolepsy and their parents frequently report memory complaints [26,27]. In a recent cross-sectional survey, Ingram and colleagues [27] reported that a difficulty in focusing and memorizing was the most frequent challenge faced by the youth and their parents. However, previous neuropsychological assessments failed to identify memory problems in patients with narcolepsy and tend to explain their difficulties in terms of attentional or executive impairments [26,28,29]. However, sleep-dependent memory consolidation, which is negatively influenced by sleep deprivation and sleep fragmentation [30], has been rarely investigated in narcoleptic patients. These studies revealed that in narcoleptic adults the sleep-dependent memory consolidation process was found to be less effective [31,32,33], suggesting that their disrupted sleep patterns hindered the complementary function of consecutive NREM-REM stages in memory consolidation [34].

### Objective and Hypothesis

This study aimed to achieve the following: (1) to investigate whether sleep transitions throughout the night and altered first REM sleep period transition (a potential EEG biomarker of narcolepsy identified in adults) are also present in narcoleptic children and (2) to explore the link between stage transitions and sleep-dependent memory consolidation in these patients. First, as in adults, we expect that the sleep stage transitions will be higher among narcoleptic children compared to control children, and the first period of REM sleep transition will be altered in narcoleptic children. Second, we hypothesize that a higher sleep transition frequency will be associated with a lower sleep-dependent memory consolidation performance.

## 2. Materials and Methods

### 2.1. Participants

Twenty-three healthy control children (mean age: 9.22 ± 1.93 years; 11 females) and twenty-one children with type 1 narcolepsy (mean age: 10.80 ± 2.27 years; 13 females) were included in the protocol. Narcoleptic children were recruited through the service of Epilepsy, Sleep and Neuropediatric Functional Explorations of the Woman Mother Child Hospital of Bron (France). Control children were recruited by the staff members of the study from those that lived in their immediate vicinity. All parents provided their written consent, and this study was approved by the local ethics committee on 1 May 2016 (n°69HCL15 0226).

For children with narcolepsy, the diagnostic procedure included sleep and wake monitoring that was conducted as follows: a clinical examination with a pediatric sleep specialist, a polysomnography from 8 p.m. to 7 a.m., and followed by 4 or 5 standard multiple sleep latency tests (MSLT) at 9 a.m., 11 a.m., 1 p.m., 3 p.m., and 5 p.m. that were ended after 20 min if no sleep occurred and after 15 min if sleep occurred [35] (see Figure 1).

Children with narcolepsy included in this study complained about excessive sleepiness for at least three months, presented clear cataplexies, and tested positive in HLA-DQB1*0602 genotyping, except for 2 children who were not tested. MSLT mean sleep latency was ≤8.2 min, and all the narcoleptic children presented two or more Sleep Onset in Rapid Eye Movement (SOREM) sleep on the MLST or during night sleep [2]. As CSF hypocretin levels were not measured in all our patients, the terminology narcolepsy with cataplexy (NC) was preferred to NT1.

NC children were de novo patients with no medication administered during the experimental protocol. Exclusion criteria were symptoms better explained by other medical or psychiatric disorders and the presence of secondary narcolepsy.

The exclusion criteria for control children were the same as for narcoleptic children. None of the participants in the control group reported any sleep complaints according to standardized sleep questionnaires (presented below).

### 2.2. Questionnaires

The Adapted Epworth Sleepiness Score (AESS), in which the item “falling asleep while in a car stopped in the traffic” was replaced with “falling asleep at school”, was used to assess daytime sleepiness [36,37]. A score over 10 was considered pathological. The Insomnia Severity Index (ISI), which contains 7 questions scored from 0 to 4, was filled out by both groups. A score greater than 10 was considered abnormal [38].

To assess symptoms of Major Depressive Disorder (MDD), we used the Children’s Depression Inventory (CDI) [39]. This questionnaire includes 27 items scored from 0 to 2. Depression was categorized by the presence and severity of symptoms with a CDI score of greater or equal to 16. The Abbreviated Conners Scale (10 items) was used to assess symptoms associated with hyperactivity and attention deficit, with a cut-off score of 15 [40].

### 2.3. Neuropsychological Assessment

Participants performed the Wechsler Intelligence Scale for Children (WISC-IV) [41]. It was not possible to calculate the total IQ for three children because they had a gap of more than 16 points between the verbal comprehension index and the perceptual reasoning index subscales. Daytime verbal and visual memory consolidation were assessed using the “Stories” and the “Dot Locations” subtests of the Children Memory Scale, respectively (CMS) [42]. During the CMS Stories subtest, children listened to short stories and were then asked to recall them exactly as presented, both immediately and after a delay of 30 min. The subtest also included a section that tested delayed recognition. Performance on the Stories subtest was measured using Immediate Recall, Delayed Recall, and Delayed Recognition subscales. The Dot Locations test evaluated children’s skills in learning and recalling an array of nine dots that are randomly positioned. It measured their performance during a learning phase and during immediate and delayed conditions.

### 2.4. Polysomnographic Recording

Polysomnographic recordings were performed according to the participants’ usual sleep schedule. Participants were instructed not to nap and not to eat/drink foods containing psychostimulants during the day preceding sleep recordings. For both groups, polysomnography included electroencephalic electrodes positioned according to the international 10–20 system (Fp1, Fp2, C3, C4, T3, T4, O1, O2), electro-oculograms, electromyogram, an oral thermistor, thoracic and abdominal belts, an electrocardiogram, a transcutaneous oximeter, and a nasal cannula. Signals were recorded and amplified using the DREAM system, Medatec, Belgium (EEG gain: 10,000; band-pass: 0.3–100 Hz; −6 dB). The oral thermistor and nasal cannula were not used in control children as these children did not present clinical signs of obstructive sleep apnea. Respiratory sensors were also removed during the multiple sleep latency test. The signal was digitized at a sampling rate of 256 Hz with Harmonie, Stellate System, Montreal, Canada. Sleep architecture, arousals from sleep, and respiratory events were visually scored in 30 s epochs by trained physicians according to the pediatric criteria of the American Academy of Sleep Medicine [35]. Total sleep time (TST), sleep efficiency, sleep and rapid eye movement (REM) latency, duration and percentage of each stage (N1, N2, N3, REM), arousal index, and wake after sleep onset (WASO) were collected. Obstructive apnea hypopnea index (OAHI), minimal oxygen saturation, index of desaturation ≥ 3%, periodic limb movement sleep index (PLMSI), and PLMSI ≥ 5/h were collected for children with narcolepsy. Twenty available MSLTs were analyzed to determine the sleep onset latency, the number of SOREM periods, and the number of stages of sleep from which REM sleep occurred within a SOREM period in NC children.

### 2.5. Stage Transitions

Overnight overall transition rate, defined as the total number of stage transition per hour of sleep between wake, N1, N2, N3 sleep, and REM sleep, was derived for each participant [43]. The overall transition rate was divided into stage-specific rates for all possible transitions. The global relative transition frequencies (%) were calculated for each stage by dividing the number of transitions between a given stage and all other stages by the total number of all transitions × 100 [44]. In addition, REM sleep transitions were specifically analyzed and the percentage of transition to REM for each sleep stage was calculated.

Because REM sleep transitions from wake or N1 during the first REM period (FREMP) and the SOREM during MSLT were previously identified as a biomarker of narcolepsy among adults [21,24], we calculated the percentage of participants entering the FREMP from each sleep stage and identified the stage preceding the SOREM of each MSLT in narcoleptic children.

### 2.6. Sleep-Dependent Memory Consolidation Task

Sleep-dependent memory consolidation was investigated using the 2D object location task inspired by the game “concentration” [45,46]. It consists of memorizing the location of pairs of cards representing colored pictures of animals or everyday objects. One hour before their habitual bedtime, participants were taken to the learning session of the task. After the learning session, a polysomnographic recording was performed during the night. In the morning, one hour after their habitual wake-up time, the recall session of the task was performed. During the learning session, randomly selected pairs of cards were shown to the children on a computer (10 pairs of cards for children aged between 6 to 8 years old, and 12 pairs of cards for the 9 to 18 years old). Children were asked to memorize the location of each pair of cards. A cued recall procedure was immediately repeated until children reached 75% of correct answers. The same cued recall procedure was performed for the recall session in the morning. The learning and recall scores (percentage of correct answers) were calculated. The overnight consolidation was measured by calculating a memory retention score: (Recall score − Learning score)/Learning score × 100. Thus, a negative memory retention score indicates a loss of information after the night, whereas a positive retention score was associated with a gain of information.

### 2.7. Statistical Analysis

We performed statistical analyses using SPSS 25.0 (SPSS Science, Chicago, IL, USA). Normality of distribution was tested using the Shapiro–Wilk test. Between group differences in demographics, sleep architecture, stage transitions, and cognitive variables were assessed using *t*-test or Chi-squares. Chi-squares were used to compare group proportion of abnormal sleep transition on FREMP. We used Pearson’s correlations to explore the link between significant sleep transitions metrics in NC children and sleep-dependent consolidation memory performance. Results are presented as mean ± standard deviation. Significance threshold was set at *p* < 0.05.

## 3. Results

### 3.1. Questionnaires and Neuropsychological Assessment

Children with narcolepsy had significantly higher scores when assessed with the sleepiness (*t*_(38)_ = −13.10; *p* < 0.001) and insomnia (*t*_(36)_ = −6.23; *p* < 0.001) questionnaires compared to control children. The mean scores of the ISI and the AESS were above the clinical cut-off for narcoleptic children (see Table 1). Although both groups had similar IQ scores, narcoleptic participants showed a significantly higher score on the verbal comprehension index (*t*_(40)_ = −2.24; *p* = 0.031).

Despite no significant difference in learning scores, we found significantly lower immediate recall (*t*_(24)_ = 2.40; *p* < 0.025) and delayed recall (*t*_(25)_ = 2.626; *p* < 0.015) for daytime visual memory performances among children with narcolepsy compared to control children, but no significant between group differences in verbal performances (stories) were observed.

### 3.2. Sleep Architecture

Narcoleptic participants exhibited shorter total sleep time (*t*_(42)_ = 2.85; *p* = 0.007) and shorter REM sleep latency (*t*_(42)_ = 5.48; *p* < 0.001) compared to participants in the control group. Sleep architecture showed a significantly higher relative time in Stage 1 (*t*_(42)_ = −2.26; *p* = 0.029) and REM sleep (*t*_(42)_ = −3.06; *p* = 0.004) for narcoleptic children, whereas control children showed significantly higher relative time in Stage 2 (*t*_(42)_ = 3.67; *p* = 0.001). Sleep was more fragmented in children with narcolepsy with a longer wake after sleep onset (*t*_(42)_ = −6.5; *p* < 0.001) and a lower sleep efficiency (*t*_(42)_ = 5.46; *p* < 0.001), for which the mean was under the threshold of 85% in NC children (83.83 ± 5.18), see Table 2.

### 3.3. Sleep Transition

#### 3.3.1. Overall and Stage-Specific Transition Rate (Number/Hour)

Narcoleptic participants exhibited significantly higher overall stage transitions per hour of sleep than participants in the control group (narcolepsy: 22.91 ± 5.62; control: 15.45 ± 3.32; *t*_(42)_ = −5.42; *p* < 0.001), see Figure 2. Table 3 presents the stage-specific rates for all possible transitions.

#### 3.3.2. Global Relative Transition Frequency

Global relative transition frequencies are presented in Table 4. Results showed significantly higher probability for NC children to shift to wake from N1 (*t*_(42)_ = −3.61; *p* < 0.001), N2 (*t*_(42)_ = −3.14; *p* = 0.003), and REM (*U* = 151.0; *p* = 0.034) than control participants. The probabilities for NC children to enter REM sleep from wake (*U* = 130.5; *p =* 0.009) or N1 (*t*_(42)_ = 2.76; *p =* 0.009) were significantly higher than control participants, who exhibited a higher probability to enter REM sleep from N2 (*t*_(42)_ = 5.48; *p* < 0.001).

Significantly lower probabilities of transition from N2 to N1 (*t*_(42)_ = 2.02 *p* = 0.05), N2 to N3 (*t*_(42)_ = 4.73; *p* < 0.05), N3 to N2 (*U* = 101.50; *p =* 0.001), and REM to N1 (*t*_(42)_ = 4.38; *p* < 0.001) were found in narcoleptic children compared to control children.

#### 3.3.3. Overnight REM Sleep Transition

Regarding the overnight transition to REM, NC children entered more frequently into REM from wake (*t*_(42)_ = −4.50; *p* < 0.001) and less frequently from N2 (*t*_(42)_ = 3.82; *p* < 0.001) than control children (see Table 5).

#### 3.3.4. Altered First REM Sleep Period Transition

Analysis of FREMPs demonstrated that 90.48% of narcoleptic children entered the FREMP from N1 or W, see Table 6. In contrast, this phenomenon was not observed in any control subjects who entered the FREMP from N2 exclusively (X^2^ = 36.2; *p* < 0.001). Moreover, as described for adults [19], the stage preceding SOREM during MSLT was always wake or N1 for NC children.

#### 3.3.5. Sleep-Dependent Memory Consolidation

As shown in Table 7, the number of trials needed to reach the minimum 75% criterion did not differ between the two groups (*t*_(42)_ = 1.27; *p* = NS), and the percentage of card pairs recalled (*t*_(42)_ = 0.39; *p* = NS) during the learning session of the 2D memory task did not differ between narcoleptic children and control children. After a night of sleep, the percentage of card pairs recalled was significantly higher in the control group compared to narcoleptic children’s group (*t*_(42)_ = 2.55; *p* = 0.015). While control children maintained their performances over the night, narcoleptic children showed a significantly lower memory retention score (*t*_(42)_ = 2.08; *p* = 0.044).

#### 3.3.6. Sleep Transitions and Sleep-Dependent Memory Performance

A negative correlation was observed between the stage transition rate and the memory retention score (*r* = −0.48, *p* = 0.013): the higher the number of transitions, the poorer the memory consolidation performance.

## 4. Discussion

We investigated whether sleep transitions throughout the night described in NT1 adults were also present in children with narcolepsy–cataplexy and explored their link with sleep-dependent memory consolidation.

As in adults, we found that NC children had significantly higher overall sleep stage transition frequency rate compared to control children. Our results are in line with what was found among adults with narcolepsy by Sorensen and colleagues [18]. We also found a higher relative sleep transition frequency to wake in all sleep stages in NC children compared to control children.

According to Sorrensen et al., nocturnal sleep instability (i.e., transitions between sleep and wakefulness) was associated with biochemical and clinical evidence of hypocretin/orexin deficiency and cataplexy [18]. Hypocretin/orexin is synthesized by a small number of neurons in the lateral hypothalamus and is circulated to various regions involved in arousal and sleep–wake regulation, including the thalamus, basal forebrain, and brainstem [47]. The loss of hypocretin/orexin neurons in narcolepsy leads to the destabilization of the flip-flop mechanism, resulting in instability between sleep and wakefulness and various symptoms of narcolepsy. The flip-flop mechanism is a model that explains the transition between wakefulness and sleep [48]. It proposes that two groups of neurons, one that promotes sleep and the other that promotes wakefulness, inhibit each other via a mutual inhibitory connection. During wakefulness, the wake-promoting neurons, i.e., hypocretin/orexin neurons, fire at high rates and inhibit the sleep-promoting neurons. Conversely, during sleep, the sleep-promoting neurons, which release gamma-aminobutyric acid from the ventrolateral and median preoptic areas of the brain, fire at high rates and inhibit the wake-promoting neurons. Fluctuations in the firing rates of these neurons lead to the switch between the two states.

Other studies on sleep stability, measured with EEG cycling alternating patterns during NREM sleep, found a lower rate of EEG synchrony among children with narcolepsy [49,50,51]. These findings are also in line with our results that show a higher overall relative sleep transition frequency that involve wakes in all NREM sleep stages. However, a lower cycling alternating pattern is not considered as a potential biomarker of narcolepsy because it is not as specific as abnormal sleep transition before FREMP in NT1.

### 4.1. Altered First REM Period Sleep Transition

As expected, the first sleep stage transition of REM sleep was altered in NC children compared to control children. As observed in previous studies, the combination of wake and N1 abnormal transition to FREMP was significantly higher among NC children. Up to 90.48% of narcoleptic children entered the FREMP from N1 or wake, whereas all control participants transitioned to FREMPs after the N2 stage. Previous studies have shown that patients with NT1 experience more awakenings and spend more time in stage N1 sleep, compared to patients with other central hypersomnias [52,53]. This observation could also apply to our results where NC children have a sleep architecture with significantly higher WASO and N1 percentage. The author explained that this difference in sleep architecture may contribute to the higher likelihood of REM sleep arising from N1 in NT1. It has been suggested that two opposing mechanisms are involved in the sleep pathology of narcolepsy. Fragmented nocturnal sleep and increased phasic REM components indicate disturbances in sleep maintenance mechanisms, while specific features observed in spectral EEG analysis suggest a decreased activity of central arousal mechanisms [54]. Our findings support the dysfunction of sleep maintenance mechanisms, leading to difficulties in progressing into deeper sleep stages and an increased pressure for REM sleep onset from lighter sleep stages in narcolepsy.

Compared to other stages, higher rates were also found for N1 or wake transition to REM in all MSLT trials with 100% of NC children entering 1st and 2nd MSLT in SOREM. Interestingly, the number of transitions from N1 to REM seemed to have increased from 65% at MSLT1 to 80% at MSLT2. However, further analyses are needed to understand whether this result is triggered by a reduction in homeostatic pressure induced by repeated naps. Indeed, releasing sleep pressure may reduce the direct transition from wakefulness to REM, allowing NC children to transition from wakefulness to N1 before transitioning to REM.

When comparing our results to studies conducted in NT1 adults, we found a higher percentage of abnormal sleep transitions on FREMPs (90.48%) and MSLT (100%) among NC children. Drakatos and colleagues found that 62.5% of adults with narcolepsy and cataplexy entered the FREMPs period from N1 or wake, while these transitions were found in 21% of narcoleptics without cataplexy and 0% of patients with other types of hypersomnia [21]. Liu et al. observed that 57.1% of NC participants entered their nocturnal FREMP from wake or N1 and 92% did so during MSLTs [24]. These results do not seem to be associated with our sample size, since they are similar to what are found in sample sizes including 24 and 35 NT1 adults, respectively. Our results oppose the hypothesis of a less severe loss of hypocretin/orexin neuron that could progress over time [17,53] and rather suggests the hypothesis that for some individuals an earlier onset of the disease could lead to a faster loss of hypocretin/orexin neurons, but this hypothesis needs to be tested.

### 4.2. Overnight REM Sleep Transitions

Until now, previous studies on NT1 abnormal sleep transitions have focused on FREMP and MSLT. No study has investigated REM sleep transitions throughout the night. Our results showed that NC children entered more frequently in REM from wake and less frequently from N2 compared to control children. Interestingly, previous research found that in individuals without narcolepsy, it is common to observe a brief transition into N2 before entering REM sleep. This pattern is less frequently observed in narcolepsy. A decreased transition from N2 to REM sleep is a specific characteristic of narcolepsy [21,24,54,55]. These results suggest that abnormal transitions previously identified before FREMPs seem to continue throughout the night.

### 4.3. Memory and Sleep Transition

As expected, we found that a higher sleep transitions rate was associated with a lower sleep-dependent memory performance. The innovative aspect of our study consists of assessing sleep-dependent memory consolidation. NC patients often report subjective memory complaints [27] and structural and functional alterations in the hippocampus of NT1 patients have been found [56]. However, very few studies that evaluated memory analyzed objective memory performance [26]. A recent study, exploring the link between subjective memory complaint and objective memory performance in narcolepsy, found similar memory performances between narcolepsy patients and participants in the control group [57]. This was possibly because it assessed memory consolidation during the day and not within a sleep period.

Our NC participants showed a decreased overnight memory performance, while control participants tended to maintain their performances. These results could not be a consequence of poor attention function in NC participants because their learning scores (before sleep) were similar to control participants. Moreover, our participants exhibited normal visuoconstructive and perceptual functioning in cognitive tests that involved these functions (Table 1). These results are compatible with the functional and structural alterations of hippocampus among type 1 narcoleptic patients as suggested by Wada et al. 2019 [56]. Sleep is known to benefit memory consolidation, but little is known about the contribution of sleep stage transition. A sequential hypothesis [58] proposes that the orderly succession of NREM than REM is necessary for memories to be replayed and integrated into existing memory networks. Recently, Strauss and colleagues [34] showed in narcoleptic adults that spindles, markers of NREM sleep, were involved in sleep-dependent memory consolidation only when learning and recall were separated by a period of NREM sleep followed by REM sleep and not when REM sleep preceded NREM sleep. Further analyses of electroencephalographic markers that have been associated with memory consolidation [59,60,61], such as slow oscillations (SO), spindles, and slow oscillation–spindle coupling, will allow us to understand the mechanisms involved in sleep-dependent memory consolidation among NT1 children.

### 4.4. Limitations

The size of our group sample is small, and hence, it could be considered as a pilot study, but it is comparable to previous samples of studies that included NT1 adults [18,21]. However, we cannot rule out the possibility that some differences in quality of life, stress, or social status may increase the differences in sleep quality observed between our two populations. This information must be studied in future studies. Previous studies used abnormal FREMP transitions to distinguish NT1 from type 2 narcolepsy and other hypersomnias, which was not performed by us. It would be beneficial to also conduct these analyses in children in future studies.

## 5. Conclusions

In this study, we found that the transitions from any sleep stage to wake were higher among NC children compared to control children. A higher sleep transition rate was associated with a lower sleep-dependent memory performance. Our study represents the first evidence of a decreased score in sleep-dependent visual memory consolidation among children with narcolepsy. The memory complaints and academic difficulties, frequently reported by narcoleptic patients, may be associated with sleep-dependent memory consolidation. Our results stress the need to investigate off-line memory consolidation in future studies and to clinically evaluate memory functioning in narcoleptic children. Altered FREMP transition, identified as an EEG biomarker of NT1 among adults, also appears to be an interesting biomarker in children. The identification of potential narcolepsy biomarkers is crucial as it may enable a quicker detection of narcolepsy in children, thereby limiting its impacts on their school, family, and social lives. Further studies with larger cohorts, including of those with type 2 narcolepsy, are needed to validate the sensitivity and specificity of this biomarker.

## Figures and Tables

**Figure 1 children-10-01702-f001:**
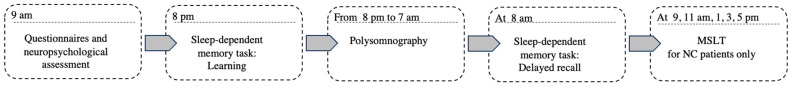
Illustration of the procedure.

**Figure 2 children-10-01702-f002:**
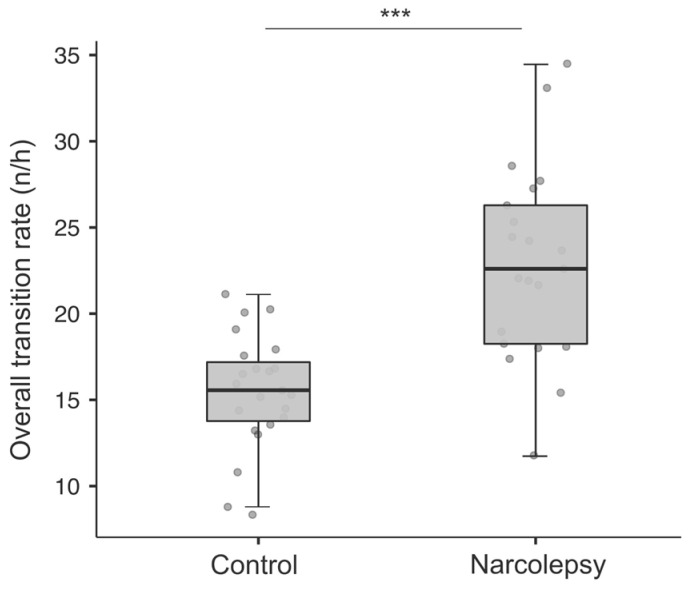
Overall stage transition rate (number of transitions per hour) in the control and narcolepsy groups. ° overall transition rate for a participant; *** *p* < 0.001.

**Table 1 children-10-01702-t001:** Questionnaires and neuropsychological assessment.

	Control Children		Narcoleptic Children			
Variables	N	Score	N	Score	t/X^2^ Values	*p* Values
Questionnaires						
AESS	21	2.86 ± 2.76	19	15.74 ± 3.45	−13.10	<0.001 **
ISI	20	4.45 ± 2.95	18	11.72 ± 4.2	−6.23	<0.001 **
CDI	19	11.81 ± 9.46	18	10.11 ± 8.22	0.59	0.57
Conners	20	5.53 ± 6.20	18	9.06 ± 7.85	−1.52	0.14
Neuropsychological assessment						
WISC-IV scores						
IQ	23	111.87 ± 12.74	16	115.88 ± 19.60	−0.77	0.44
VCI	23	116.78 ± 12.68	19	126.84 ± 16.40	−2.24	0.031 *
PRI	23	109.39 ± 8.86	19	109.16 ± 16.40	0.06	0.953
WMI	23	102.61 ± 12.58	19	106.74 ± 12.30	−1.07	0.292
PSI	23	103.04 ± 19.88	18	99.61 ± 2.10	0.55	0.589
Daytime verbal memory: Stories						
Immediate recall	15	12.67 ± 2.32	11	11.82 ± 2.34	0.766	0.451
Delayed recall	15	12.47 ± 2.47	11	12 ± 3.19	0.42	0.678
Delayed recognition	14	13.21 ± 2.46	9	11.33 ± 2.65	1.741	0.096
Daytime visual memory: Dots localization						
Learning	14	11.79 ± 1.89	12	10.25 ± 3.05	1.569	0.13
Learning and immediate recall	14	12.00 ± 0.39	12	10.00 ± 3.10	2.397	0.025 *
Delayed recall	15	12.53 ± 2.07	12	10.25 ± 2.45	2.626	0.015 *

Notes: Results are presented as mean ± standard deviation. AESS: Adapted Epworth Sleepiness Scale; ISI: Insomnia Severity Index; CDI: Children’s Depression Inventory; Conners: Abbreviated Conners Scale; IQ: Intellectual quotient, VCI: verbal comprehension index; PRI: perceptual reasoning index; WMI: working memory index; PSI: processing speed index. * *p* < 0.05; ** *p* < 0.01.

**Table 2 children-10-01702-t002:** Sleep architecture.

	Control Children		Narcoleptic Children			
Variables	N	Score	N	Score	t/X^2^ Values	*p* Values
TST (min)	23	531.57 ± 45.68	21	481.38 ± 45.68	2.85	0.007 **
LAT (min)	23	29.98 ± 30.73	21	5.14 ± 5.61	3.67	0.001 **
Latency REM (min)	23	145.7 ± 59.06	21	43.19 ± 64.1	5.48	<0.001 **
Sleep efficiency (%)	23	92.62 ± 5.48	21	83.83 ± 5.18	5.46	<0.001 **
N1 (min)	23	63.87 ± 24.34	21	74.68 ± 31.69	−1.28	0.209
N1 (%)	23	11.97 ± 3.89	21	15.61 ± 6.57	−2.26	0.029 *
N2 (min)	23	238.83 ± 45.79	21	185.07 ± 32.92	4.43	<0.001 **
N2 (%)	23	44.65 ± 5.38	21	38.43 ± 5.87	3.67	0.001 **
N3 (min)	23	123.61 ± 28.39	21	104.48 ± 25.17	2.36	0.023 *
N3 (%)	23	23.54 ± 5.73	21	21.76 ± 5.24	1.08	0.288
REM (min)	23	104.26 ± 28.34	21	117.12 ± 28.53	−1.5	0.141
REM (%)	23	19.67 ± 4.98	21	24.2 ± 4.81	−3.06	0.004 **
Arousals (nb)	23	10.49 ± 3.76	21	13.92 ± 5.08	−2.50	0.017 *
WASO (min)	23	33.97 ± 28.55	21	90.64 ± 29.25	−6.5	<0.001 **
AHOI, n/h			21	0.62 ± 0.05		

Notes: Results are presented as mean ± standard deviation; TST: total sleep time; LAT: sleep latency; min: minutes; N1: NREM stage 1; N2: NREM stage 2; N3: NREM stage 3; REM: rapid eye movement; WASO: wake after sleep onset; AHOI = apnea–hypopnea obstructive index * *p* < 0.05; ** *p* < 0.01.

**Table 3 children-10-01702-t003:** Stage-specific transition rates.

Control Children					
(nb/h)	W	N1	N2	N3	REM
W		1.97 ± 0.85	0.65 ± 0.41	0.08 ± 0.10	0.35 ± 0.26
N1	0.75 ± 0.49		2.77 ± 0.90	0.06 ± 0.08	1.13 ± 0.61
N2	1.18 ± 0.61	1.65 ± 0.95		1.16 ± 0.57	0.45 ± 0.20
N3	0.36 ± 0.17	0.16 ± 0.31	0.78 ± 0.61		0.00 ± 0.00
REM	0.70 ± 0.43	1.06 ± 0.54	0.19 ± 0.20	0.00 ± 0.00	
**Narcoleptic children**					
**(nb/h)**	**W**	**N1**	**N2**	**N3**	**REM**
W		4.40 ± 1.83 ***	1.09 ± 0.63 **	0.18 ± 0.19 *	1.06 ± 0.60 ***
N1	1.99 ± 1.09 ***		3.98 ± 1.90 **	0.03 ± 0.06	1.10 ± 0.51
N2	2.61 ± 0.91 **	1.83 ± 1.15		0.86 ± 0.41	0.28 ± 0.23 *
N3	0.67 ± 0.29 **	0.09 ± 0.16	0.31 ± 0.30 **		0.00 ± 0.00
REM	1.51 ± 0.65 ***	0.71 ± 0.51 *	0.21 ± 0.21	0.00 ± 0.00	

Notes: Stage-specific transition rates (n/h). Mean ± standard deviation between subjects in the number of transitions per hour of sleep from sleep stage in the leftmost column to sleep stage in the topmost row. W: wake; N1: NREM stage 1; N2: NREM stage 2; N3: NREM stage 3; REM: rapid eye movement. * *p* < 0.05; ** *p* < 0.01; *** *p* < 0.001.

**Table 4 children-10-01702-t004:** Global relative transition frequencies between stages.

Control Children					
(%)	Wake	N1	N2	N3	REM
W		12.87 ± 4,83	4.17 ± 2.33	0.53 ± 0.71	2.57 ± 3.32
N1	4.69 ± 2.84		17.87 ± 3.74	0.35 ± 0.51	7.26 ± 3.48
N2	7.86 ± 3.88	10.21 ± 4.89		7.48 ± 3.43	2.97 ± 1.15
N3	2.44 ± 1.43	1.03 ± 1.92	4.85 ± 3.88		0.00 ± 0.00
REM	4.77 ± 3.08	6.84 ± 3.29	1.23 ± 1.27	0.00 ± 0.00	
**Narcoleptic children**					
**(%)**	**Wake**	**N1**	**N2**	**N3**	**REM**
W		18.64 ± 4.98 ***	5.36 ± 4.74	0.88 ± 1.02	5.21 ± 3.97 **
N1	8.53 ± 4.15 ***		16.41 ± 5.84	0.10 ± 0.24	4.85 ± 2.04 **
N2	11.70 ± 4.23 **	7.48 ± 3.94 *		3.72 ± 1.30 ***	1.20 ± 0.97 ***
N3	3.05 ± 1.46	0.37 ± 0.64	1.29 ± 1.02 **		0.0 ± 0.0
REM	7.05 ± 3.89 *	3.06 ± 2.31 ***	1.09 ± 1.25	0.0 ± 0.0	

Notes: Global relative transition frequencies (%) between sleep stages. Mean ± standard deviation between subjects in the global relative transition probability from sleep stage in the leftmost column to sleep stage in the topmost row. W: wake; N1: NREM stage 1; N2: NREM stage 2; N3: NREM stage 3; REM: rapid eye movement. * *p* < 0.05; ** *p* < 0.01; *** *p* < 0.001.

**Table 5 children-10-01702-t005:** Characterization of the transition to REM.

	Control Children	Narcoleptic Children
Stage preceding REM stage (%)		
W	18.92 ± 13.71	42.09 ± 20.07 ***
N1	55.75 ± 18.49	46.21 ± 20.68
N2	25.33 ± 13.08	11.71 ± 10.22 ***
N3	0.00 ± 0.00	0.00 ± 0.00
Number of participants entering the first REM sleep period (FREMP) from different stages		
W	0/23 (0%)	2/21 (9.52%)
N1	0/23 (0%)	17/21 (80.96%)
N2	20/23 (86.96%)	1/21 (4.76%)
N3	0/23 (0%)	0/21 (0%)
Unknown	3/23 (13.04%)	1/21 (4.76%)
W	0/23 (0%)	2/21 (9.52%)

Notes: Results are presented as mean (%) ± standard deviation and number of children in each group; W: wake; N1: NREM stage 1; N2: NREM stage 2; N3: NREM stage 3; REM: rapid eye movement. *** *p* < 0.001.

**Table 6 children-10-01702-t006:** Stages preceding SOREM during MSLT.

	MSLT1	MSLT2	MSLT3	MSLT4	MSLT5
No. of patients with SOREM					
	20/20 (100%)	20/20 (100%)	16/20 (80%)	13/20 (65%)	6/7 (85.71%)
Stage preceding SOREM					
W	7/20 (35.0%)	4/20 (20%)	3/16 (18.75%)	2/13 (15.38%)	1/6 (16.67%)
N1	13/20 (65.0%)	16/20 (80%)	13/16 (81.25%)	11/13 (84.62%)	5/6 (83.33%)
N2	0/21 (0%)	0/20(0%)	0/16 (0%)	0/13 (0%)	0/6 (0%)
N3	0/21 (0%)	0/20(0%)	0/16 (0%)	0/13 (0%)	0/6 (0%)

Notes: Results are presented in number and percentage of NC children. MSLT: multiple sleep latency test; SOREM: presence of REM periods within the 15 min after sleep onset; W: wake; N1: stage N1; N2: stage N2; N3: stage N3.

**Table 7 children-10-01702-t007:** Performance in the 2D object location task.

	Control Children		Narcoleptic Children			
Variables	N	Score	N	Score	t/X^2^ Values	*p* Values
Learning phase (before PSG recording)						
Number of trials	23	3.48 ± 1.93	21	2.86 ± 1.20	1.27	0.211
Correct answers (%)	23	82.96 ± 8.37	21	82.05 ± 6.88	0.39	0.698
Recall session (after PSG recording)						
Correct answers (%)	23	84.26 ± 12.84	21	74.00 ± 13.85	2.55	0.015 *
Memory retention score	23	2.09 ± 16.40	21	−9.10 ± 12.25	2.08	0.044 *

Notes: Results are presented as mean ± standard deviation; * *p* < 0.05.

## Data Availability

The data presented in this study are available on request from the corresponding author.

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
