# Peer review of "Sleep Stage Transitions and Sleep-Dependent Memory Consolidation in Children with Narcolepsy–Cataplexy"

_children, 2023, doi:10.3390/children10101702_

Round 1

Reviewer 1 Report

Of particular relevance is the study of problems of memory impairment during sleep in children. This is important as it can greatly affect children's ability to learn in school. Therefore, the research conducted by the authors is undoubtedly relevant and interesting.

The article is well structured, the manuscript is written at a good methodological level. During the review, a number of comments and additions arose:

1) Authors need to carefully check the article for punctuation errors and inaccuracies (for example, there is no period in line No. 80);

2) The research design is presented in continuous text, for clarity, it is necessary to build a block diagram;

3) It would be appropriate to display the comparative characteristics of study groups in the form of a table. Including information related to the assessment of the social status of volunteers (parental family or single-parent family, quality of life, stress, daily routine, additional activities outside of school), all this can also affect sleep. There is also no data on what time the subjects usually go to bed and what time they wake up, how many hours their usual sleep lasts, or whether they have daytime sleep (how many hours). Family history is also interesting, whether close relatives had sleep problems. The subjects' diet (consumption of caffeine-containing products), frequency and timing of consumption of these products are also important.

4) The list of references contains 64 sources, but no references to 52 and 55 sources of literature were found in the text. Authors must carefully check the text of the article for the presence of all listed sources of literature from the list.

Author Response

Reviewer 1

1) Authors need to carefully check the article for punctuation errors and inaccuracies (for example, there is no period in line No. 80).

We agree with the reviewer, the article has been revised for punctuation accuracies.

2) The research design is presented in continuous text, for clarity, it is necessary to build a block diagram.

We add a diagram of the study protocol to the paper, please see Page 3.

Figure 1. Illustration of the procedure

3) It would be appropriate to display the comparative characteristics of study groups in the form of a table. Including information related to the assessment of the social status of volunteers (parental family or single-parent family, quality of life, stress, daily routine, additional activities outside of school), all this can also affect sleep. There is also no data on what time the subjects usually go to bed and what time they wake up, how many hours their usual sleep lasts, or whether they have daytime sleep (how many hours). Family history is also interesting, whether close relatives had sleep problems. The subjects' diet (consumption of caffeine-containing products), frequency and timing of consumption of these products are also important.

We agree with the reviewer that many characteristics such as the social status and family history of sleep issues can impact sleep quality and would have been interesting to add to our experimental design. Unfortunately, required information is missing, which is now a limitation addressed in the manuscript as follow:

Page 12, lines 432-434. “We cannot rule out the possibility that some differences in quality of life, stress or social status may increase the differences in sleep quality observed between our two populations. This information will have to be studied in future studies”.

To ensure that our results were not related to a change in sleep schedule, the participants were equipped and installed in their recording room in the evening, but the time at which they fell asleep was not imposed. Patients were also asked to refrain from napping during the day before their sleep and nap recording and to limit consumption of psychostimulants such as caffeinated soft drinks. The same instructions were applied to the control subjects. These elements were added to the methodology.

Page 4, lines 159-161. “Polysomnographic recordings were performed according to the participants' usual sleep schedule. Participants were instructed not to nap and to eat/drink foods containing psychostimulants during the day preceding sleep recordings.”

4) The list of references contains 64 sources, but no references to 52 and 55 sources of literature were found in the text. Authors must carefully check the text of the article for the presence of all listed sources of literature from the list.

We thank the reviewer for noticing this mistake. The literature references have been updated both in the text and the Reference section.

Reviewer 2 Report

"Sleep stage transitions and sleep-dependent memory consolidation in children narcolepsy-cataplexy" is an interesting study, but

- the sample size is very small - so it is only a pilot study

- abstract & conclusion are too short

- linguistic style has to be improved

"Sleep stage transitions and sleep-dependent memory consolidation in children narcolepsy-cataplexy" is an interesting study, but

- linguistic style has to be improved - by a native speaker.

Author Response

Reviewer 2

"Sleep stage transitions and sleep-dependent memory consolidation in children narcolepsy-cataplexy" is an interesting study, but

- the sample size is very small - so it is only a pilot study

We agree with the reviewer that our sample size is small and mention it in the limitation of our study line 468. However, we do not agree that this is only a pilot study because our sample is composed of well identified children with narcolepsy without any medication, or other medical issue. Moreover, previous studies on narcolepsy included the same sample size of narcolepsy cataplexy in adults and were not considered as pilot studies.

- abstract & conclusion are too short

Abstract and conclusion have been completed, please see pages 1 and 12.

- linguistic style has to be improved

English has been corrected.

Reviewer 3 Report

This is an interesting study. Some minor suggestions

Please justify why was Wechsler Intelligence Scale for Children (WISC-IV) was chosen.

Please elaborate on this: All the sleep stages were visually scored in 30-second epochs by trained physicians according to the pediatric criteria of the American Academy of Sleep Medicine.

Include a literature search of studies looking at the outcomes in children with narcolepsy and memory/intelligence

How this study adds to the current knowledge and future recommendations

Author Response

Reviewer 3

This is an interesting study. Some minor suggestions

Please justify why was Wechsler Intelligence Scale for Children (WISC-IV) was chosen.

We thank the reviewer for this comment. Wechsler Intelligence Scale for Children 4th edition was the scale available at the time of the experiment in the hospital department where children with narcolepsy were recruited.

Please elaborate on this: All the sleep stages were visually scored in 30-second epochs by trained physicians according to the pediatric criteria of the American Academy of Sleep Medicine.

We understand that the sentence was ambiguous. The scoring rules for sleep architecture and arousals from sleep were scored using the AASM criteria (Iber, 2007), which are the same for both the pediatric and adult age groups. However, the American Academy of Sleep Medicine recommends specific pediatric rules for scoring respiratory events in children aged < 13 years. Those rules were applied in our sample. The sentence has been modified as follows:

Page 4, lines 170-172: “Sleep architecture, arousals from sleep and respiratory events were visually scored in 30-second epochs by trained physicians according to the pediatric criteria of the American Academy of Sleep Medicine [42].”

Include a literature search of studies looking at the outcomes in children with narcolepsy and memory/intelligence

We have added with the references concerning memory a reference concerning the intellectual efficiency of narcoleptic children. This study showed that children with narcolepsy exhibit normal intellectual development.

Page 2, lines 76-78: “While their intellectual efficiency meets and occasionally exceeds the standard [25], children with narcolepsy and their parents frequently report memory complaints [26-27]”.

[25] Thieux M, Zhang M, Marcastel A, Herbillon V, Guignard-Perret A, Seugnet L, Lin JS, Guyon A, Plancoulaine S, Franco P. Intellectual Abilities of Children with Narcolepsy. J Clin Med. 2020 Dec 17;9(12):4075. doi: 10.3390/jcm9124075. PMID: 33348677; PMCID: PMC7766444.

How this study adds to the current knowledge and future recommendations

We thank the reviewer for this comment. The conclusion has been modified according to reviewers’ comments.

Page 12, lines 442-452: “Our study represents the first evidence of a decreased score in sleep-dependent visual memory consolidation among children with narcolepsy. The memory complaints and academic difficulties frequently reported by narcoleptic patients may be associated to sleep-dependent memory consolidation. Our results stress the need to investigate off-line memory consolidation in future research and clinical evaluation of memory functioning in those children. Altered FREMPs transition, identified as an EEG biomarker of NT1 among adults, also appears to be an interesting marker in children. The identification of potential biomarkers is crucial as it may enable quicker detection of narcolepsy in children, thereby limiting its impacts on their school, family, and social lives. Further research with larger cohorts including type 2 narcolepsy is needed to validate the sensitivity and specificity of this biomarker.  .” 

Round 2

Reviewer 2 Report

"Sleep stage transitions and sleep-dependent memory consolidation in children narcolepsy-cataplexy" is mostly corrected in sense of reviewers.

But the sample size is very small - so it is in my opinion only a pilot study !

"Sleep stage transitions and sleep-dependent memory consolidation in children narcolepsy-cataplexy" :

- linguistic style has been improved

Author Response

We took the reviewer's opinion into account and added the following sentence in the limitations section of our study (line 431) 

"The size of our group sample is small and cloud be considered as a pilot study, but it is comparable to previous samples in studies including NT1 adults"
